# Hazards of Radioactive Mineralization Associated with Pegmatites Used as Decorative and Building Material

**DOI:** 10.3390/ma15031224

**Published:** 2022-02-06

**Authors:** Mohamed M. El Dabe, A. M. Ismail, Mohamed Metwaly, Sherif A. Taalab, Mohamed Y. Hanfi, Antoaneta Ene

**Affiliations:** 1Nuclear Materials Authority, Maadi, Cairo 11381, Egypt; Mohamedeldabe@yahoo.com (M.M.E.D.); a.m.ismail302@gmail.com (A.M.I.); dr_mohamedmetwaly@yahoo.com (M.M.); 2Faculty of Science, Al-Azhar University, Cairo 11884, Egypt; sheriftaalab@azhar.edu.eg; 3Institute of Physics and Technology, Ural Federal University, St. Mira, 19, 620002 Yekaterinburg, Russia; 4INPOLDE Research Center, Department of Chemistry, Physics and Environment, Faculty of Sciences and Environment, Dunarea de Jos University of Galati, 47 Domneasca Street, 800008 Galati, Romania

**Keywords:** Gabal El Urf, pegmatitic rocks, zoned pegmatite, poly-phased mineralization uranothorite, radiological hazard parameters, radioactive minerals

## Abstract

The present study aimed to assess the radiological hazards associated with applying the investigated granite in the building materials and the infrastructures applications. The investigated granites are classified into four categories: El-Urf, barren, colourful and opaque. El Urf monzogranite intrudes metagabbro diorite complex with sharp contacts. Based on the activity concentrations, the environmental parameters such as absorbed dose rate (D_air_), annual effective dose (AED), radium equivalent activity (Ra_eq_), external (H_ex_) and internal (H_in_) hazard indices were measured. The mineralized pegmatite is located in the southwestern foothill of the Gabal El Urf younger granite. It displays well-defined zonation of three zones: outer, middle and inner zones represented by potash feldspar, quartz and mica, respectively. The isorad map showed that El Urf monzogranite is barren (Up to 100 cps) surrounding an excavation of the studied pegmatite that exhibits moderate colorful mineralization (phase-I = 500–1500 cps) and anomalous opaque mineralization (phase-II = 1500–3500 cps) pegmatites. The obtained results of radionuclides activity concentrations illustrated that the Opaque granites have the highest values of ^238^U (561 ± 127 Bq kg^−1^), ^232^Th (4289 ± 891 Bq kg^−1^) and ^40^K (3002 ± 446 Bq kg^−1^) in the granites, which are higher than the recommended worldwide average. Many of the radiological hazard parameters were lesser than the international limits in the younger granites and barren pegmatites. All of these parameters were higher in the colorful and opaque mineralized pegmatites. The high activity and the elevated radiological hazard parameters in the mineralized pegmatites are revised to the presence of radioactive and radioelements bearing minerals, such as thorite, meta-autunite, kasolite, phurcalite, columbite, fergusonite, Xenotime and fluorapatite. Other instances of mineralization were also recorded as cassiterite, atacamite, galena, pyrite and iron oxide minerals. Thus, the granites with high radioactivity concentration cannot be applied in the different applications of building materials and ornamental stones.

## 1. Introduction

Granites are igneous rocks generally made up of quartz, K-feldspar and mica, and are used for internal and exterior decorative uses, including building and ornamental materials. Because of their nature, these rocks contain radionuclides. Exposure to the radioactive series ^238^U and ^232^Th, as well as ^40^K, produces external irradiation. Internal doses from radon inhalation and the aforementioned radioactive chains’ short-lived products are concentrated in respiratory tract tissues [1,2,3]. Uranium (U) and thorium (Th) series of natural radionuclides can be found in various levels in all terrestrial materials, depending on the geological and geographical conditions of the study area [4,5]. They can be found in almost every environment and can even be identified in the human body [3]. The terrestrial radionuclides and their daughters and cosmic radiation contribute to background radiation in the environment. Mineralogical, geochemical and physicochemical factors all play a role in its presence in the environment [6,7]. In recent years, there has been a lot of discussion about the radiological risk posed by building materials [6]. 

Moreover, the radiological impact of the general public is a major topic of research in radioecology, where the data will provide importantly and required information in monitoring environmental contamination, allowing the public to access more appropriate and effective protection advice [8,9]. The production of gamma radiation from natural radionuclides must be closely monitored in order to safeguard humans against gamma radiation, which can be caused by various diseases [10,11]. According to the ATSDR (Agency for Toxic Substances and Disease Registry), long-term radioactive exposure causes significant ailments that include oral necrosis, chronic lung disease, leukopenia and anaemia [12,13]. Several studies have been carried out to estimate the radiation risk and yearly dose supply of natural radioactivity in building materials [14,15]. Implementing a radiological impact assessment for construction materials in order to analyze and control radioactive consequences on humans and the environment is a critical and complex task that must be carried out in order to meet the criteria for sustainable development. Radiation effects should be assessed using quantifiable values that can be utilized as input parameters for designing environmental distribution and estimating radiation dose [16,17]. The present paper concerns the geological and mineralogical composition of the studied mineralized pegmatite and its environmental impacts on humans and the environment. Some of the radiological risks such as radium equivalent activity (Ra_eq_), absorbed dose rate (D_air_), annual effective dose (AED), external (H_ex_) and internal (H_in_) hazard indices and gamma index (Iγ) are computed. 

## 2. Materials and Methods

### 2.1. Geological Setting

The studied mineralized pegmatite located in the eastern part of Gabal El Urf granite, Central Eastern Desert of Egypt, and bounded by latitudes 26°37′58″–26°38′11″ N and longitudes 33°26′51″–33°28′09″ E (Figure 1).

Gabal El Urf younger granite has an elongate shape, nearly striking NE–SW. It is monzogranite with medium to coarse-grains, and has calc-alkaline to alkaline nature affinity, with a nearly estimated Sr-Nd age of 600 ± 11 Ma [18,19]. Many pegmatite bodies and masses had intruded the metagabbro diorite complex, representing El Urf monzogranite’s country rocks [20]. Many pegmatite bodies and masses had intruded the metagabbro diorite complex, representing El Urf monzogranite’s country rocks [21]. They display zoned pegmatites constituting a source for the mineralization of radioactive and rare metals (Y, Th, Nb, and Zr) [22]. The pegmatites derived from metaluminous to peralkaline magma fall within the plate granite type and are enriched with cheralite (Ca-rich monazite) and zircon [23]. Hydrothermal processes are enriched with rare metals mineralization and radioactive minerals [21]. In general, all pegmatite rocks in Gabal El Urf younger granite have been recorded in their country rocks. The latter revealed that a huge zoned mineralized pegmatite body had intruded the El Urf younger granite, with an average of nearly 14×7m in size. It is characterized by potash feldspar, quartz and mica minerals, and outer, middle and inner zones. They (Optic.) recoded earlier colorful mineralization (phase-I) and latter opaque stages (phase-II). It can be documented that the main difference between the two mineralized phases is attributable to time gapping, not the spatial issue. Both colourful and opaque mineralization stages can be found in the same location in the pegmatite zones. However, obviously, the opaque minerals phase-II (latter) cut the earlier colourful phase-I, indicated by both field investigations as well petrographic studies. El Urf monzogranite intrudes metagabbro diorite complex with sharp contacts [18,21]. The studied mineralized pegmatite is located in the southwestern foothill of the Gabal El Urf younger granite. It displays well definite zonation and consists of three zones: outer, middle and inner zones represented by potash feldspar, quartz and mica, respectively. Generally, the huge studied pegmatite had been noticed by the diggers who look after the potash feldspar masses. They excavate all the masses they can find, which are used in the ceramic industry. After excavating and removing quartz pockets, some unexposed potash masses had appeared, in which some radioactive minerals were contained (Figure 2). 

The studied mineralization includes both colorful (phase-I) and opaque mineralization (phase-II); rarely they occur consistently in the same place (Figure 3). Generally, the colorful phase-I occurs as clots of disseminated minute crystals with bright colors ranging from yellow to green in quartz and potash feldspar (Figure 4). Opaque mineralization phase-I displays as a network of fracture-filling iron associated with a vast array of accessory minerals (Figure 5). Opaque mineralization (phase-II) displays a coarser grain size of minerals than (phase-I). It includes iron oxides and mega crystals of colorless and purple fluorite associated with black radioactive minerals (Figure 6); purple fluorite is an indication to the radioactive influence. 

According to the petrographic and mineralogical studies, Phase-I encloses accessory minerals such as thorite, fluorite, zircon and xenotime, whereas the latter (phase-II) has another array of the accessory minerals, for example, fluorapatite, cassiterite, atacamite, Nb-minerals and sulfide minerals, besides the Th-minerals (thorite, uranothorite,), U-minerals (meta-autunite and uranophane) and REE-bearing minerals (pyrochlore and bastnasite), associated mainly with fractures filled by iron oxides. 

Gabal El Urf monzogranite is bounded from the south by the elliptical pegmatitic body that distinguished the moderate radioactive pegmatite phase-I surrounded by the anomalous radioactive pegmatite phase-II (Figure 7). In addition, structurally, both the colorful mineralization phase-I and opaque phase-II are mainly located near or along with definite fractures, leading to easy migration or removal of uranium ions, especially at the oxidizing regime.

### 2.2. Radiometric and Mineral Analysis 

The radiometric field survey of the El Urf younger granite and its related pegmatite was carried out using the portable scintillometer (UG-130), measuring in terms of count per second (Cps), and also determined as equivalent uranium (eU), thorium (eTh) and potassium (K). Before the measurements were carried out in the field, the portable scintillator was calibrated using the calibration pads which are certified by IAEA. The calibration experiment was designed by Matolin’s (1990) [24]. The obtained data by UG-130 were in agreement with the NaI (Tl) detector. A Nickon polarized microscope (Olympus-BZ70) mainly examined the petrographic studies to recognize the radioactive minerals and radioelement-bearing minerals of the studied mineralized pegmatite. The X-ray diffraction technique (XRD), using a Philips PW 3710/31 diffractometer, scintillation counter, Cr & Cu target tube and Ni filter at 40 kV and 30 mA. This instrument is connected to a computer system using the APD program and PDF-2 database for mineral identification. An scanning electron microscope (SEM model Philips XL30) supported by an energy dispersive spectrometer (EDX) unit was used at 25-30 kV accelerating voltage, 1–2 mm beam diameter and 60–120 s counting time. All the analyses were carried out at the labs of the Nuclear Materials Authority (NMA), Cairo, Egypt. Table 1 summarizes how to calculate radiological risk factors using activity concentrations of ^238^U, ^232^Th and ^40^K, and the mathematical equations.

## 3. Results and Discussion

### 3.1. Mineralogical Studies

The mineralogical studies of the pegmatite rocks of Gabal El Urf were carried out to determine the minerals that cause radioactive anomalies and identify the minerals that contain rare earth elements associated with uranium and thorium elements present in the two mineralized phase-I and phase-II. In phase-I sections, radioactivity refers to the mica minerals that include an array of radioelement-bearing minerals in addition to the presence of xenotime, zircon and fluorite. Phase-II mineralized pegmatite spots are characterized by an array of significant minerals comprising thorite, meta-autunite, kasolite and phurcalite. Nb-minerals include columbite and fergusonite, xenotime, fluorapatite, cassiterite, atacamite, sulfide minerals galena and pyrite, and iron oxide minerals.

#### 3.1.1. Thorite ((Th, U) SiO_4_)

Thorite mineral is presented in phase-I as minute grains included in the mica minerals and as a fracture filling. El Dabe (2022) illustrated the EDX analysis of thorite minerals containing thorium (35.64%) and uranium (10.51%), representing the main constituents with the silicate (10.94%) [31]. Yttrium is the sole trace element (5.34%) occupying a limited percentage of U-cote, according to the similarity of the ionic radii (Figure 8). Thorite exists in phase-II relatively more than in phase-I. It is presented in phase-II as small grains, disseminated clusters and microfracture filling. The XRD analysis shows that the d-spacing of Thorite characteristic peaks (3.56, 4.71, and 2.65 A^O^) at 2theta (24.98, 18.79, and 33.78), respectively, matches PDF-2 Card No. (11–17) (Figure 9). Its composition is confirmed by EDX analysis, where thorium (61.14) and uranium (13.98) represent the main constituents, along with silica (13.74) and Yttrium (8.43) (Figure 8). It is clear that thorium and uranium in phase-II exceed the corresponding one in phase-I. This coincides with the entire measuring e(Th) and e(U) values belonging to both (phase-I) and (phase-II) pegmatite parts. 

#### 3.1.2. Phurcalite Ca_2_(UO_2_)_3_(PO_4_)_2_(OH)_4_·4H_2_O

Phurcalite is a secondary calcium uranium phosphate mineral product of hydrothermal activity and is presented in (phase II) mineralized spots of pegmatites. The XRD pattern (Figure 10) shows the d-spacing of its characteristic peaks (8.00, 3.09, and 2.88 A^O^) at 2theta (16.45, 43.45 and 46.79), respectively, and matches PDF-2 Card No. (30-284).

#### 3.1.3. Meta-Autunite (Ca(UO_2_)_2_(PO_4_)_2_·6H_2_O)

Uranyl phosphate minerals are present in phase-II as Meta-autunite minerals formed by the dehydration of autunite that crystallized from the hydrous magma. An XRD investigation analysis was carried out. The XRD pattern shows that the d-spacing of its characteristic peaks (8.62, 3.48, and 3.66 A^O^) at 2theta (15.27, 38.39, and 36.37), respectively, matches PDF-2 Card No. (28-247). (Figure 11).

#### 3.1.4. Kasolite Pb(UO_2_)SiO_4_·H_2_O

Kasolite is the uranyl silicate mineral riches by lead. It is presented in the (phase-II) mineralization of pegmatites. The XRD pattern of kasolite shows that the d-spacing of its characteristic peaks (2.92, 3.07, and 3.26 A^O^) at 2theta (30.58, 29.06, and 27.30), respectively, matches PDF-2 Card No. (29-788) (Figure 12).

#### 3.1.5. Xenotime (YPO_4_)

Xenotime is an yttrium phosphate presented in phase-I. The mineral is enriched in rare earth elements (REE) and is associated with thorium and uranium minerals. The EDX-analysis indicates the presence of appreciable contents of Th (8.84%) and U (2.47%). The XRD pattern of Xenotime shows that the d-spacing of its characteristic peaks (3.45, 2.56, and 1.76 A^O^) matches PDF-2 Card No. (83-658) (Figure 13).

### 3.2. Radioactivity and Radiological Hazards Assessment

Generally, the difference among uranium and thorium contents, as well as their ratio values of El Urf monzogranite, barren pegmatite and mineralized pegmatite phase-I and phase-II, suggest the removal or migration concept of uranium ions from the whole pegmatite parts by different manners. Two uranium migration styles were revalued. The colorful mineralization (phase-I) with the barren parts migrated its uranium content through the regular method, whereas the uranium ion charge was removed from opaque mineralization (phase-II) and the El Urf monzogranite due to its disturbance and irregular style. Table 2 displays the radiometric data to clarify the distribution of radioactivity and locate the three levels of radioactivity. The obtained results showed that El Urf monzogranite is barren (Up to 100 cps), surrounding an excavation of the studied pegmatite that exhibits moderate colorful mineralization (phase-I = 500–1500 cps) and anomalous opaque mineralization (phase-II = 1500–3500 cps) of pegmatites. 

The eU and eTh contents values in ppm, as well as K, in %, were converted to activity concentration, Bq kg^−1^, using the conversion factors (12.35 and 4.06 Bq kg^−1^/ppm for ^238^U and ^232^Th, respectively, as well as 313 Bq kg^−1^/ % for ^40^K) [32], where A_U_, A_Th_ and A_K_ are the average activity concentrations of ^238^U, ^232^Th and ^40^K in Bq kg^−1^, respectively. The ^238^U activity concentration ranges between 16 and 46, averaging 34 Bq kg^−1^, and ^232^Th activity concentration varies between 16 and 45 Bq kg^−1^ with 30 Bq kg^−1^ as an average. ^40^K ranges between 406.90 and 1201.92 Bq kg^−1^ with an average of 914 Bq kg^−1^. The studied El Urf younger samples have slightly lower activity than the worldwide average values for ^238^U and ^232^Th but with higher amounts of ^40^K. The worldwide average values are 33, 45 and 412 Bq kg^−1^ for ^238^U, ^232^Th and ^40^K, respectively [3]. Activity concentrations of ^238^U, ^232^Th and ^40^K in the barren pegmatites range from (64 to 123), (69 to 89) and (1083 to 2075) with averages (93, 78, 1568), respectively. All the barren pegmatite samples have activity concentrations higher than the permissible levels, (Table 1). The radioactive pegmatites are classified according to the opaque mineral contents into colorful mineralized pegmatite phase and opaque mineralized phase. Activity concentrations ranges of ^238^U, ^232^Th and ^40^K in the colorful pegmatite phase are (208–499), (566–1854) and (1561–3449), with averages of 321, 991.66 and 2354, respectively, whereas for opaque mineralized pegmatite, phases are (364–763), (2799–5733) and (2212–3506), with averages of 561, 4289 and 3002, respectively. The activity concentrations averages of both colorful and opaque mineralized pegmatites are much higher than the international averages mentioned later (Table 1).

#### Radiological Hazards Parameters

The mean Ra_eq_ values for the granitic rocks of El Urf are 147, 324, 1919 and 6919 Bq kg^−1^ for the younger granite, barren pegmatite and colorful and opaque mineralized pegmatites, respectively. However, colorful and opaque mineralized pegmatites have much higher values than the criterion limit of 370 Bq kg^−1^; however, the younger granite and barren pegmatite are lower. However, colorful and opaque mineralized pegmatites display much higher values than the criterion limit of 370 Bq kg^−1^, whereas the younger granite and barren pegmatite are lower (Figure 14).

These indices must be less than the average in order to keep the radiation hazard insignificant [33,34,35]. The radiation exposure due to radioactivity in construction materials must be limited to 1.5 mSv year^−1^ (Table 2). The values of the external hazard (H_ex_) and internal hazard (H_in_) for the studied granitic rocks are less than standard in El Urf younger granites, which agrees with the recommended values, whereas almost samples in the barren pegmatite and all samples in the colorful and opaque mineralized pegmatites are, to a greater extent, higher than international standards, suggesting that these samples cannot be used as a building or decorative material of dwelling (Table 2). The opaque mineralized pegmatites parts have the both the highest external hazard (H_ex_) and internal hazard (H_in_) values, reflecting the most dangerous used material among the studied rock types in the El Urf area (Figure 15). The safety value for this index is ≤1, whereas the obtained Iγ averages for the studied rocks are 1.13, 2.44, 13.63 and 48.63 for younger granites, barren pegmatite and colorful and opaque mineralized pegmatites, respectively. Most of the studied rocks have a value higher than the recommended safety value, to a great extent. 

Table clarifies the estimated gamma-absorbed dose rate values for the studied granitic rock samples. The D_air_ values for the younger granite samples range from 34 to 99 nGy h^−1^, with a mean of 72 nGy h^−1^. Barren pegmatite samples range from 118 to 187 nGy h^−1^, with a mean of 99 nGy h^−1^. D_air_ values for both colorful and opaque mineralized pegmatites are (502–1495) and (1948–3954), with averages of 845 and 2975, respectively. The mean D_air_ values for all the studied granitic rocks exceed the worldwide average value (59 nGy h^−1^, UNSCEAR, 2000), (Table 2). This displays that the Gabal El Ur area is not appropriate for the stratification of various infrastructure applications, particularly building materials.

The mean values of the studied granitic rocks are 0.09, 0.19, 1.04 and 3.65, for the younger granite and barren pegmatite, colorful and opaque mineralized pegmatites, respectively, which are higher than the recommended worldwide average of the annual effective dose (0.07 mSvy^−1^), as suggested by UNSCEAR (2000) [36], (Table 2). Heavy minerals found in granites, such as monazite, uraninite and thorianite, are responsible for the high doses. Furthermore, long-term exposure to high dosages might have negative health consequences such as cancer and cardiovascular disease, which are linked to tissue degradation and deoxyribonucleic acid (DNA) in genes [37].

The principal component analysis (PCA) employed Varimax rotations to identify the matrix connection between distinct components. The PC1 and PC2 components are shown in Figure 16.

In opaque granite samples, the activity concentrations of ^238^U and ^232^Th indicate a strong positive in PC1 loading, which is linked to all radiological factors and explains 98.94% of the variation. As a result, ^238^U and ^232^Th activity concentrations were the most common natural radioactive contributions in the opaque granite at the research location. PC2 accounts for 0.92 % of the variance [38,39]. 

The data of radiological variables are analyzed using a hierarchical clustering approach. Figure 17 depicts the relationship between all of the variables. The dendrogram of the examined data in the opaque granite at the El Urf area shows two clusters. Cluster I in the opaque granite at the analyzed location is made up of ^238^U, ^232^Th and radiological hazard factors. Although cluster II contains the ^40^K, which are linked to cluster I, this analysis demonstrated that uranium and thorium minerals are responsible for the total radioactivity in the opaque granite. Finally, the cluster analysis results are consistent with PCA analysis.

## 4. Conclusions

The mineralized pegmatite is located in the southwestern foot hill of the Gabal El Urf younger granite and displays well-defined zonation of three zones: outer, middle and inner zones represented by potash feldspar, quartz and mica, respectively. The activity concentrations of ^238^U, ^232^Th and ^40^K in the mineralized pegmatitites have higher values relative to the worldwide average. The highest values of ^238^U (561±127 Bq kg^−1^), ^232^Th (4289±891 Bq kg^−1^) and ^40^K (3002 ± 446 Bq kg^−1^) are found in the opaque mineralized pegmatites. Many of the radiological hazard parameters were lesser than the international limits in the younger granites and barren pegmatites. All these parameters were higher in the colorful and opaque mineralized pegmatites. This is attributed to the alteration of radioactive minerals such as radioactive earing minerals such as thorite, meta-autunite, kasolite, phurcalite, columbite, fergusonite, xenotime and fluorapatite. Other instances of mineralization were also recorded as cassiterite, atacamite, galena, pyrite and iron oxide minerals. The statistical analysis was conducted to illustrate the geological processes that lead to an increase in the radioactive concentration in the granite rocks. Thus, the granite rocks of the studied area are not safe, pose negative health risks and are not applied in the building materials and the application of various infrastructures.

## Figures and Tables

**Figure 1 materials-15-01224-f001:**
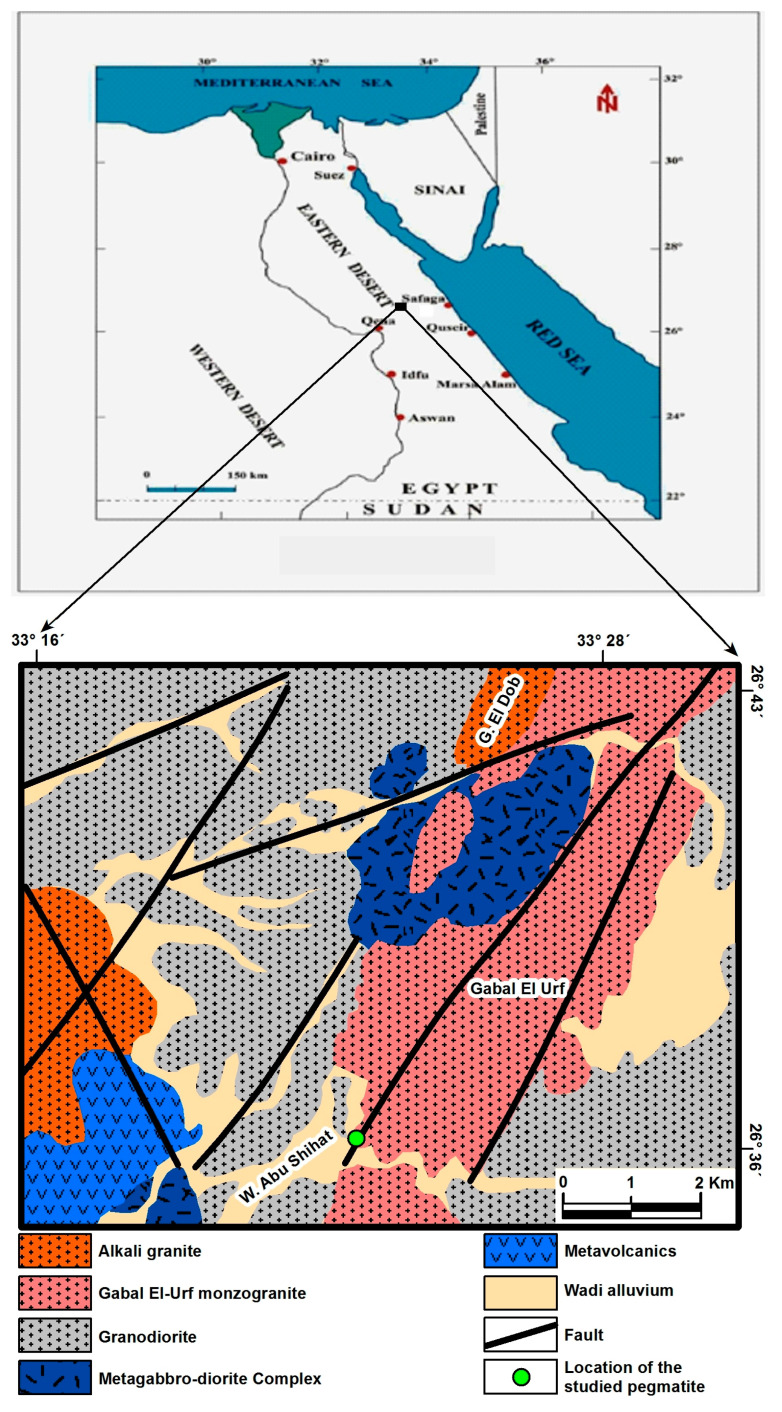
Location map of Gabal El Urf younger granite, Central Eastern Desert, Egypt. [18].

**Figure 2 materials-15-01224-f002:**
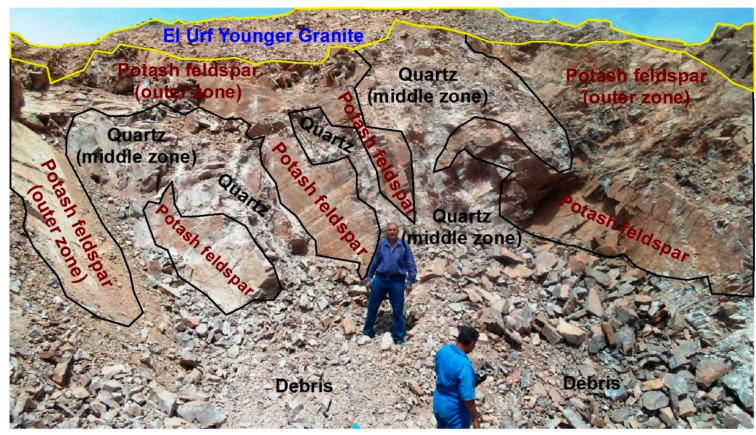
A close view of the western part of the huge studied mineralized pegmatite shows newly potash exposure masses containing opaque mineralization after excavation in the mineralized pegmatite of Gabal El Urf area, looking NE.

**Figure 3 materials-15-01224-f003:**
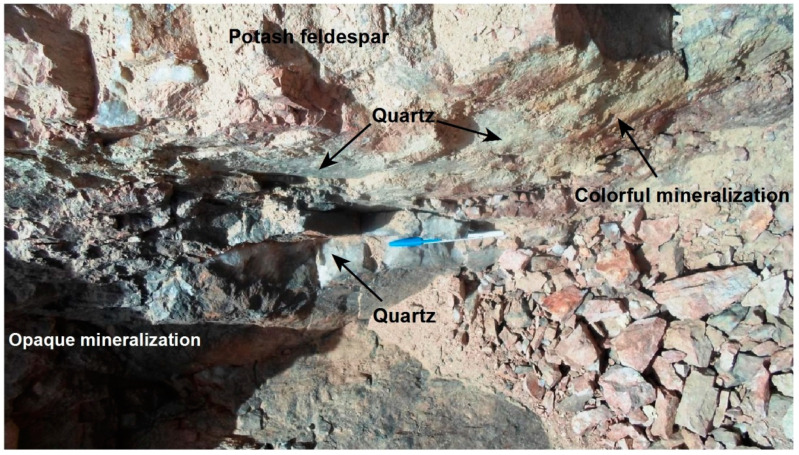
Both the colorful phase-I and opaque phase-II mineralization between the outer and middle zones, in the mineralized pegmatite of Gabal El Urf area, looking NE.

**Figure 4 materials-15-01224-f004:**
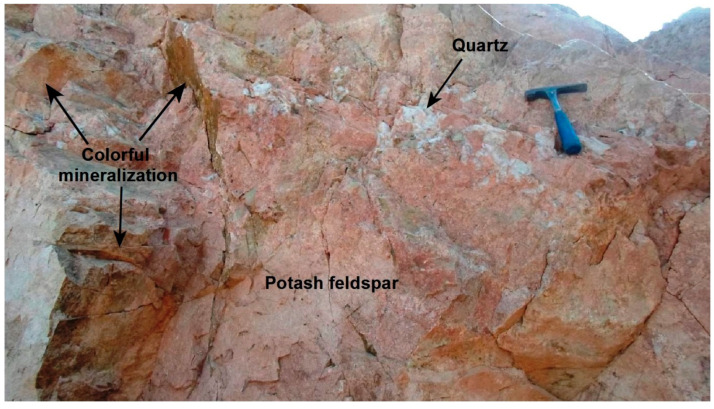
A close view of the colorful mineralization (phase-I) disseminated in potash feldspar associated with a quartz dyke-like body, looking N.

**Figure 5 materials-15-01224-f005:**
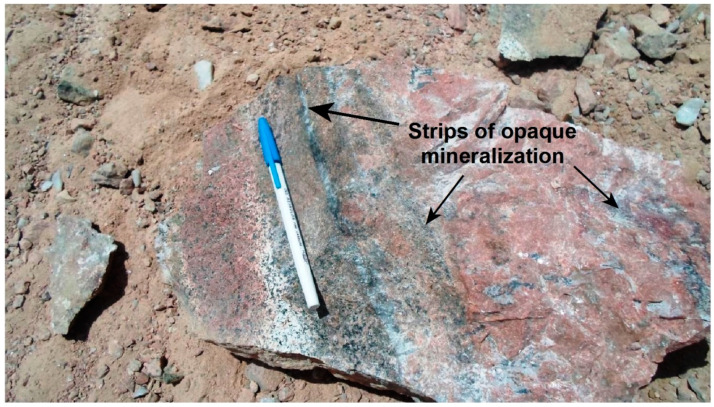
A hand specimen showing parallel bands of iron oxides with high radioactivity minerals (phase-II), associated with quartz veinlets.

**Figure 6 materials-15-01224-f006:**
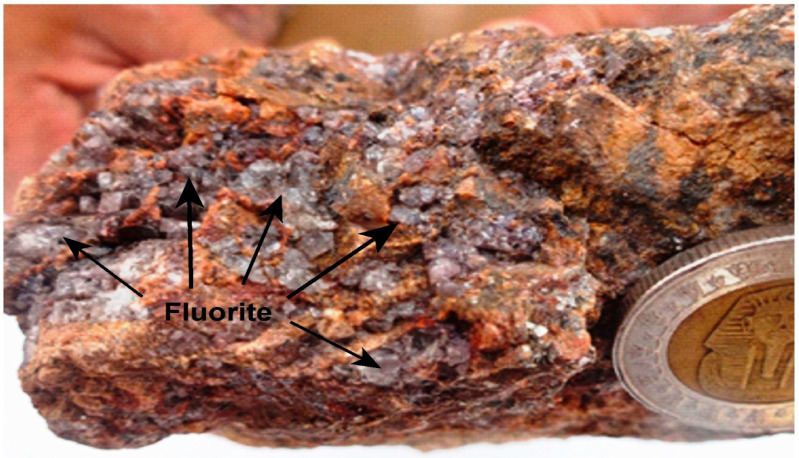
A hand specimen showing iron oxides associated with megacrystals of colorless and purple fluorite with high radioactivity minerals (phase-II).

**Figure 7 materials-15-01224-f007:**
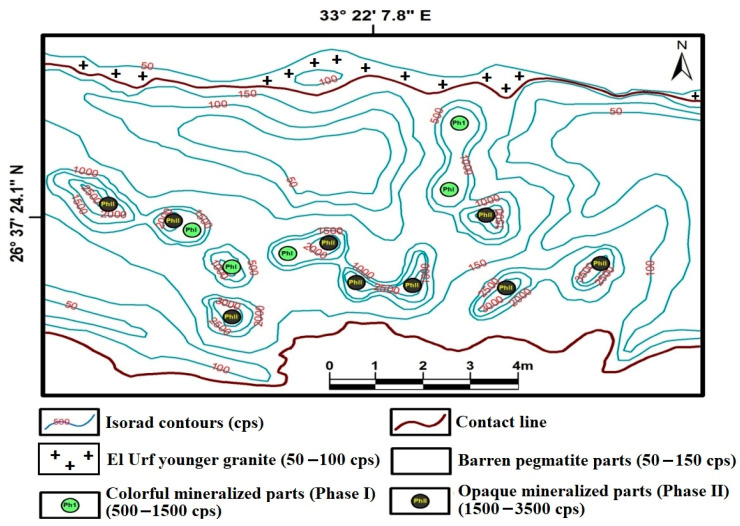
Radiometric map showing the distribution of radioactive measuring values of the two phases of mineralization.

**Figure 8 materials-15-01224-f008:**
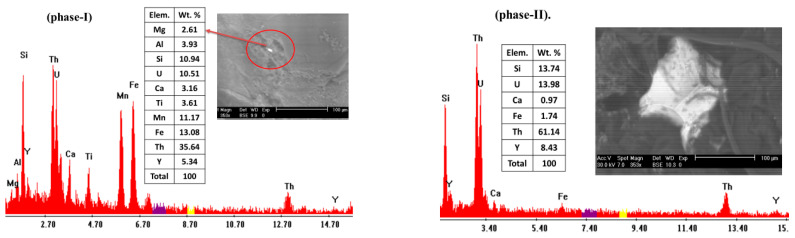
EDX-analysis of thorite, El-Urf mineralized pegmatite (phase-I), and (phase-II).

**Figure 9 materials-15-01224-f009:**
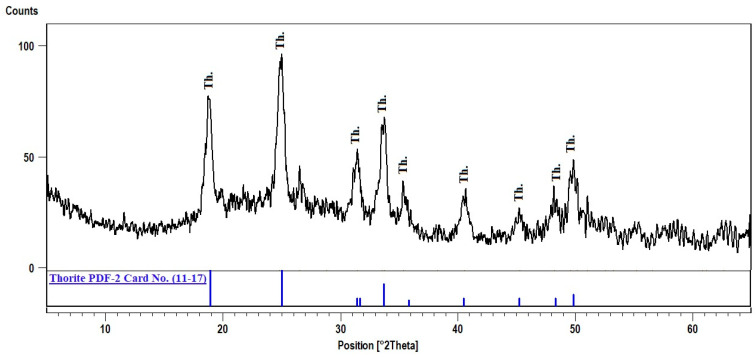
XRD pattern of thorite, El Urf mineralized pegmatite (phase-II).

**Figure 10 materials-15-01224-f010:**
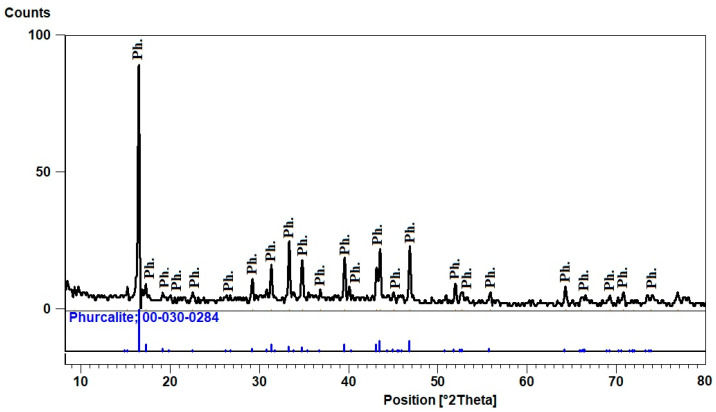
XRD pattern of Phurcalite, El Urf mineralized pegmatite (phase-II).

**Figure 11 materials-15-01224-f011:**
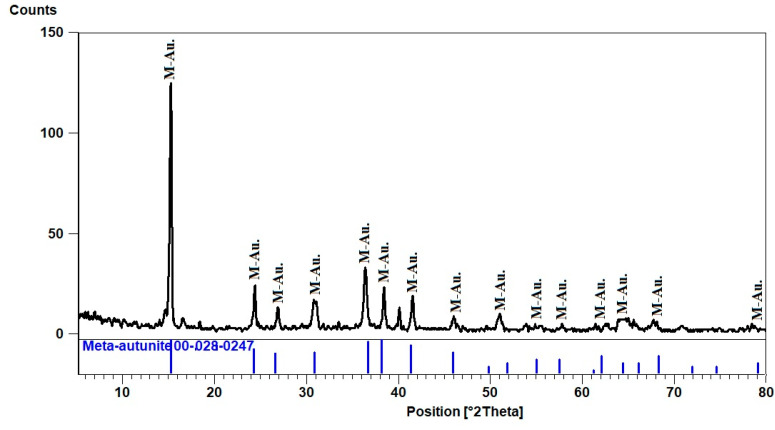
XRD pattern of Meta-autunite, El Urf mineralized pegmatite (phase-II).

**Figure 12 materials-15-01224-f012:**
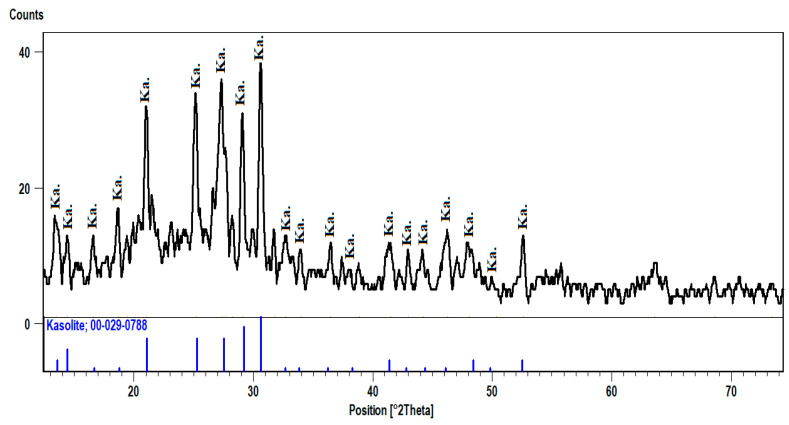
XRD pattern of Kasolite, El Urf mineralized pegmatite (phase-II).

**Figure 13 materials-15-01224-f013:**
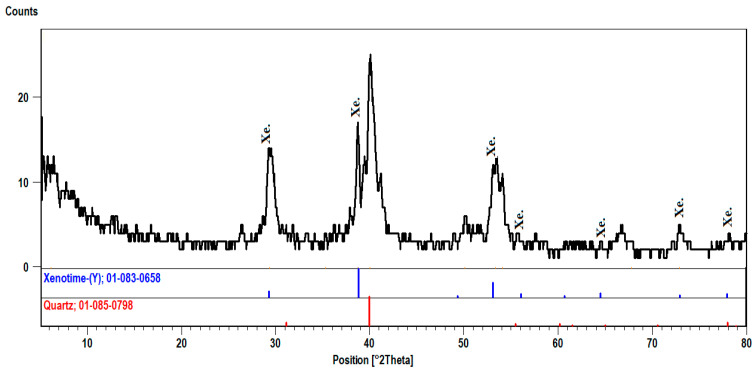
XRD pattern of xenotime, El-Urf mineralized pegmatite parts (phase-I).

**Figure 14 materials-15-01224-f014:**
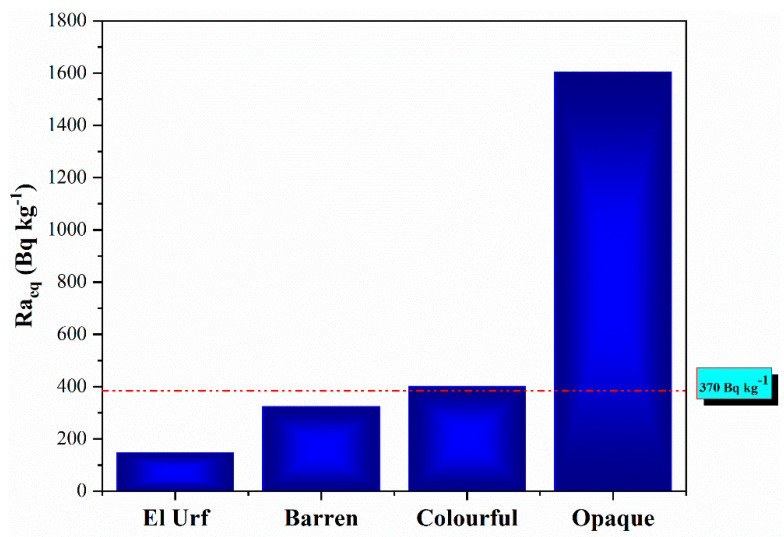
An illustrating histogram shows the average radium equivalent activity (Raeq) values for the El Urf younger granite and different mineralized pegmatite parts.

**Figure 15 materials-15-01224-f015:**
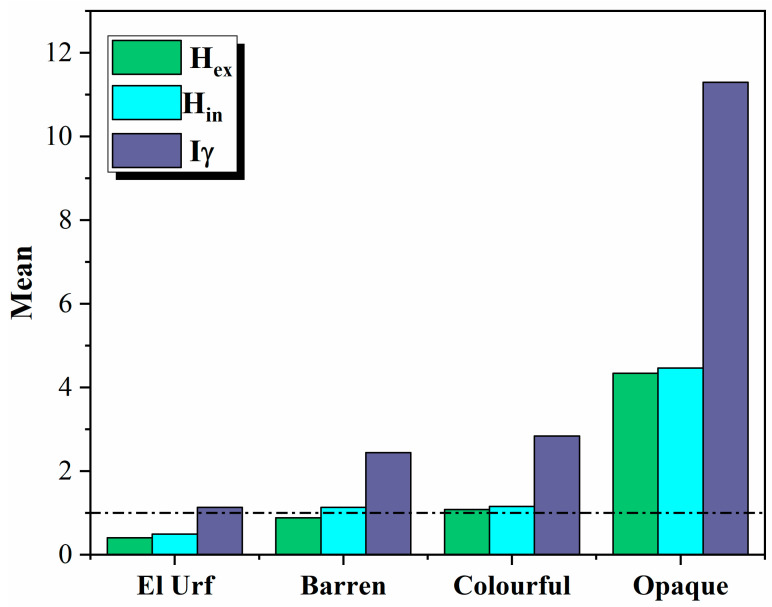
An illustrating histogram shows the average external hazard values (H_ex_), internal hazard (H_in_) and radiation level index (Iγ) for the El Urf younger granite and different mineralized pegmatite parts.

**Figure 16 materials-15-01224-f016:**
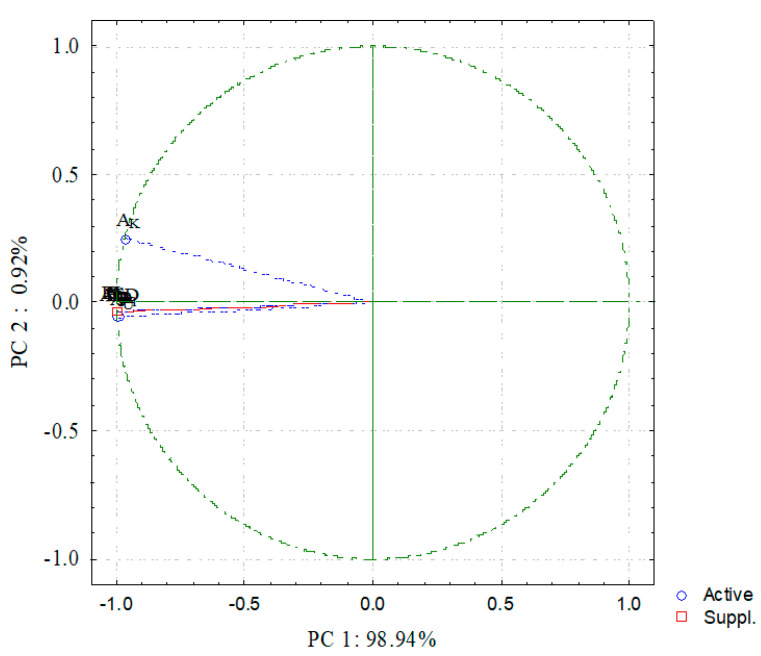
Principal component analysis (PC1 and PC2) for radiological data of opaque granite at the El Urf area.

**Figure 17 materials-15-01224-f017:**
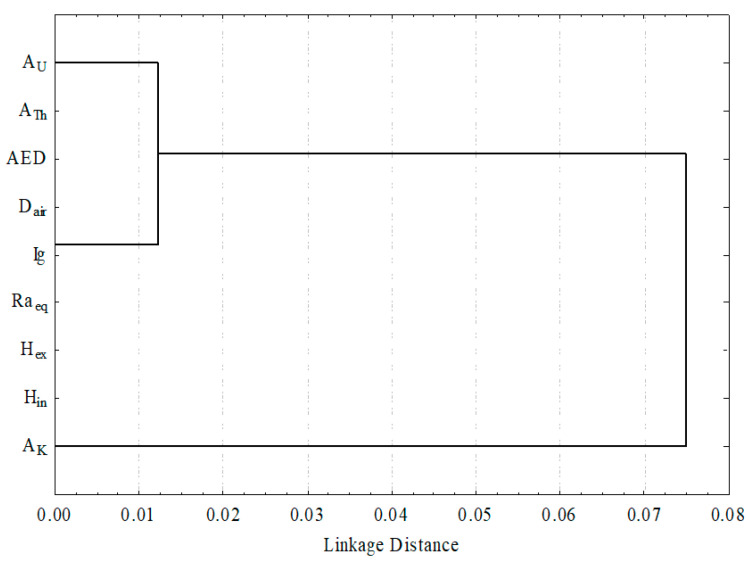
The clustering analysis of the radiological parameters of Opaque at EL Urf area.

**Table 1 materials-15-01224-t001:** Important radiological parameters and indices.

Parameter	Symbol	Definition	Formula
Radium equivalent activity	Ra_eq_	Radium equivalent activity is a weighted sum of the ^226^Ra, ^232^Th and ^40^K activities according to the hypothesis that 370 Bq kg^−1^ of ^226^Ra, 259 Bq/kg of ^232^Th and 4810 Bq/kg of ^40^K attain the same dose rates of gamma rays	Ra_eq_ (Bq kg^−1^) = A_Ra_ + 1.43 A_Th_ + 0.077 A_K_
External hazard index	H_ex_	The external hazard index is the radiological parameters applied to assess the hazard of γ-radiation [25,26]	$H_{\mathrm{ex}}=\frac{A_{U}}{370}$ $+\frac{A_{\mathrm{Th}}}{259}$ $++\frac{A_{K}}{4810}$
Internal hazard index	H_in_	The internal hazard index is applied to the internal exposure from radon and its decay products [27,28]	$H_{\mathrm{in}}=\frac{A_{U}}{185}$ $+\frac{A_{\mathrm{Th}}}{259}$ $++\frac{A_{K}}{4810}$
Radiation level index	Iγ	The other index used to estimate the level of γ-radiation hazard associated with the natural radionuclides in the samples was suggested by a group of experts due to the different combinations of specific natural activities in the sample [29,30]	$I_{\gamma }=\frac{A_{\mathrm{Ra}}}{150}$ $+\frac{A_{\mathrm{Th}}}{100}$ $++\frac{A_{K}}{1500}$
Absorbed dose rate	D (nGy/h)	The absorbed dose rate is the radioactive factor that was applied to detect the effect of gamma radiation at 1 m from the radiation sources in the air due to the concentrations of ^238^U, ^232^Th and ^40^K	D_air_ (nGy h^−1^) = 0.430 A_U_ + 0.666 A_Th_ + 0.042 A_K_
Outdoor annual effective dose	AED	The annual effective dose is a radioactive factor utilized to detect the exposure level for radiation during a stationary duration (1 year)	AED (mSv y^-1^) = D_air_ (nGy/h) × 0.2 × 8760 (h/y) × 0.7 (Sv/Gy) × 10^−6^ (mSv/nGy)

**Table 2 materials-15-01224-t002:** Results of radionuclide activity concentrations, the dose rate (D_air_), the annual effective dose (AED), radium equivalent activity (Ra_eq_), external (H_ex_), internal (H_in_) hazard indices and gamma index (Iγ) for younger granite and barren pegmatite samples.

Rock Type	Series of Observations No #.	A_U_	A_Th_	A_K_	Dair	AED	Ra_eq_	H_ex_	H_in_	I_γ_
		Bq kg^−1^	Bq kg^−1^	Bq kg^−1^	(nGyh^−1^)	mSv	Bq kg^−1^			
El Urf younger granite	1	25	23	664	53.31	0.07	109	0.30	0.36	0.84
	2	16	16	407	33.85	0.04	70	0.19	0.23	0.54
	3	42	37	1099	87.65	0.11	180	0.49	0.6	1.38
	4	40	28	898	72.63	0.09	149	0.4	0.51	1.14
	5	46	45	1202	98.79	0.12	203	0.55	0.67	1.56
	Av	34	30	914	71.75	0.09	147	0.4	0.49	1.13
	SD	11	10	291	23	0.03	48	0.13	0.16	0.37
	min	16	16	407	33.85	0.04	70	0.19	0.23	0.54
	max	46	45	1202	98.79	0.12	203	0.55	0.67	1.56
Barren pegmatite parts	6	64	71	1083	117.75	0.14	249	0.67	0.85	1.86
	7	87	82	1474	151.26	0.19	318	0.86	1.09	2.38
	8	92	86	1552	158.85	0.19	333	0.9	1.15	2.5
	9	119	69	2009	180.23	0.22	372	1	1.33	2.82
	10	84	74	1424	142.87	0.18	299	0.81	1.04	2.25
	11	79	84	1340	142.97	0.18	302	0.82	1.03	2.26
	12	123	69	2075	184.98	0.23	381	1.03	1.36	2.89
	13	96	89	1621	165.46	0.2	347	0.94	1.2	2.61
	14	121	75	2050	186.85	0.23	386	1.04	1.37	2.93
	15	85	74	1434	143.95	0.18	301	0.81	1.04	2.27
	16	75	77	1277	134.29	0.16	283	0.76	0.97	2.12
	17	87	82	1474	150.95	0.19	317	0.86	1.09	2.38
	Av	93	78	1568	155.01	0.19	324	0.88	1.13	2.44
	SD	18	6	305	20	0.03	40	0.11	0.16	0.31
	min	64	69	1083	117.75	0.14	249	0.67	0.85	1.86
	max	123	89	2075	186.85	0.23	386	1.04	1.37	2.93
Colorful Mineralized parts (phase-I)	18	267	743	1969	132.22	0.16	305	0.82	0.88	2.16
	19	208	566	1562	101.01	0.12	233	0.63	0.67	1.65
	20	403	1327	2980	230.17	0.28	533	1.44	1.53	3.77
	21	282	791	2100	140.57	0.17	324	0.88	0.94	2.3
	22	500	1854	3449	316.7	0.39	734	1.99	2.09	5.19
	23	301	851	2225	151.01	0.19	348	0.94	1.01	2.47
	24	288	809	2197	143.66	0.18	331	0.9	0.96	2.35
	Av	321	992	2354	173.62	0.21	401	1.08	1.15	2.84
	SD	91	413	594	69	0.09	160	0.43	0.45	1.13
	min	208	566	1562	101.01	0.12	233	0.63	0.67	1.65
	max	500	1854	3449	316.7	0.39	734	1.99	2.09	5.19
Opaque Mineralized parts (phase-II)	25	365	2866	2194	457.27	0.56	1071	2.89	2.97	7.53
	26	700	5315	3409	849.92	1.04	1990	5.38	5.53	14
	27	508	3650	2936	585.77	0.72	1371	3.7	3.82	9.65
	28	444	3366	2570	538.35	0.66	1260	3.41	3.5	8.87
	29	365	2799	2132	447.29	0.55	1047	2.83	2.91	7.37
	30	565	4291	3271	686.2	0.84	1607	4.34	4.47	11.3
	31	683	5187	3446	829.35	1.02	1942	5.25	5.4	13.66
	32	544	4150	3155	663.33	0.81	1553	4.2	4.32	10.93
	33	764	5398	3506	867.37	1.06	2029	5.48	5.65	14.28
	34	612	4767	3205	760.95	0.93	1782	4.82	4.95	12.54
	35	365	2821	2113	450.51	0.55	1055	2.85	2.93	7.42
	36	569	4750	3296	755.14	0.93	1769	4.78	4.91	12.44
	37	754	5733	3440	916.67	1.12	2146	5.8	5.96	15.1
	38	631	4954	3393	790.43	0.97	1851	5	5.14	13.02
	Av	561	4289	3002	685.54	0.84	1605.05	4.34	4.46	11.29
	SD	127	891	446	143	0.18	335	0.90	0.93	2.36
	min	365	2799	2113	447.29	0.55	1047	2.83	2.91	7.37
	max	764	5733	3506	916.67	1.12	2146	5.8	5.96	15.1

## Data Availability

The data presented in this study are available on request from the corresponding authors.

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
