# Peer review of "Hazards of Radioactive Mineralization Associated with Pegmatites Used as Decorative and Building Material"

_materials, 2022, doi:10.3390/ma15031224_

Round 1
Reviewer 1 Report
The paper is quite interesting and seems to me worth of publishing. However some corrections and clarification are needed. As I don’t feel myself competent enough regarding the geological aspects, my remarks will be only limited to the radiation protection and radiation measurements issues.
In the attached file a list of comments, remarks and requests for clarification.
Good luck!

Author Response
Dear Reviewer,
Please find attached the submission of the carefully revised version of the manuscript in Ref., following the minor comments and modification of the Reviewer.
Below is a detailed list of the changes made in response to the Reviewer’s minor comments (in italics), which outlines every change made a point by point. The changes are marked in the manuscript text (yellow highlighted).
The paper is quite interesting and seems to me worth of publishing. However some corrections and clarification are needed. As I don’t feel myself competent enough regarding the geological aspects, my remarks will be only limited to the radiation protection and radiation measurements issues.
In the following a list of comments, remarks and requests for clarification
1) Abstract, line 18 …specific activity concentration: please delete the word specific, in order to avoid confusion. The specific activity, referred to a radionuclide, is the activity per unit mass of the radionuclide itself, i.e.: As=decay constant * Avogadro number / Atomic mass number
Response: The word specific deleted from the manuscript.
2) Introduction, line 6 …produce the external dosage. Replace with: … produce external irradiation
Response: Done.
3) Introduction, line 9 …, are the opposite: are opposite to what ? not clear, please clarify
Response: It was a mistake, so it deleted from the sentence.
4) Line 23-25 Not really true. The exposure to gamma radiation is not specifically linked to lung cancer: the association to this disease, if any, is weak. Lung cancer is related to radon inhalation. Of course the detection of gamma radiation from the natural radionuclides producing radon can be used to estimate the radon exhalation and the related risk. Please modify the sentence.
Response: Of course, you are right. It was a mistake. Therefore, all information corrected in the manuscript.
5) Page 12, Paragraph 3.2, lines 17-22 Activity concentrations…..ppm. As long as they are expressed in ppm the concentrations of the elements are not “activity concentrations” but simply concentration. Please delete Activity. …where AU, ATh, AK are the average specific activities…: please replace with “the average activity concentrations”, see point 1)
Response: Corrected in the manuscript and explained in detail. The radioactive content of U, Th, and K in ppm is converted to the activity concentration in Bq kg-1 using the conversions factors.
6) Page 13, Paragraph 3.2, line 3 …the world concentration limits…: as far as I know there is no “world concentration limits”. Please clarify and explain what do you mean.
Response: Corrected in the manuscript. The authors mean the worldwide average values of 238U and 232Th and 40K are 33, 45 and 412, respectively, (UNSCEAR, 2000).
7) Page 13, Paragraph 3.2, lines 22-36 Absorbed gamma dose rate This section needs a general clarification. It is not clear to me if the calculated Absorbed gamma dose rate whose values are reported in table 1 are referred to 1 meter above the ground, accordingly with the cited UNSCEAR formula, or are the indoor gamma dose rate for a standard house constructed using the materials under study. Please note the indoor gamma dose rate cannot be calculated simply using the outdoor UNSCEAR formula. This section must be rewritten indicating clearly the calculation methods used for the estimation of the indoor gamma dose rate.
Response: Corrected and explained in the manuscript. Yes, of course, I agree with your opinion, where the indoor is not included in the current study. Thus, the annual effective dose in the investigation is related to the outdoor only.
8) Page 13, Paragraph 3.2, line 45-46 …the permissible annual effective dose. Please indicate the value of this dose in the text
Response: Corrected and explained in the manuscript. The values are related to the worldwide average (0.07 mSv y-1).
9) Page 16, Paragraph 3.2, line 3 …specific activities respectively…Please substitute the strikethrough words with: activity concentrations
Response: Done in the manuscript.
10) Page 16, Paragraph 3.2, line 13-15 Please indicate in the references the original work from which the formulas Hex and Hin were taken. The cited references are referred to works that simply used such formulas.
Response: The original references are added to the manuscript.
11) Please provide also the original reference for the gamma index Ig, reported in Table 1, page 14.
Response: The original reference was added to the manuscript.
We thank the Reviewer a lot for the useful and valuable comments that have helped improve the manuscript.
Hoping that all the careful review is sufficient for the direct acceptance of the manuscript, thank you for your time and consideration.
Best wishes,
Mohamed. Y. M. Hanfi
on behalf of all co-authors
Reviewer 2 Report
The paper Hazards of radioactive mineralization associated with pegmatites used as decorative and building material presents results of the assessment of radiological hazards associated with applying the investigated granite in the building materials and the infrastructures applications.
I find that the paper is not suitable for publication in the special issue: Future Trends in Advanced Materials and Processes.
One of the reasons is the paper brings no new knowledge in the field of radioactivity of NORM.
Also, the equipment used is not adequate for precise measurements of radioactivity (the usual methodology of measuring radioactivity using gamma-ray spectrometry should be applied, or some other precise measurements. The calculations and conversion of ppm-s to Bq/kg is not adequate.
Additionally, the results of this study are only off local interest.
Author Response
Dear Reviewer,
Please find attached the submission of the carefully revised version of the manuscript in Ref., following the minor comments and modification of the Reviewer.
Below is a detailed list of the changes made in response to the Reviewer’s minor comments (in italics), which outlines every change made a point by point. The changes are marked in the manuscript text (turquoise highlighted).
The paper Hazards of radioactive mineralization associated with pegmatites used as decorative and building material presents results of the assessment of radiological hazards associated with applying the investigated granite in the building materials and the infrastructures applications.
- I find that the paper is not suitable for publication in the special issue: Future Trends in Advanced Materials and Processes.
Response: Before the submission, the authors scanned a journal, “Materials”, and found some similar publications in this issue of the journal “Future Trends in Advanced Materials and Processes.” Therefore, the authors believe the paper is appropriate for the journal.
- One of the reasons is the paper brings no new knowledge in the field of radioactivity of NORM.
Response: The present study aimed to assess the radioactivity in the granite, which can be extracted from various areas in Egypt. The investigated granites can be applied in various infrastructures fields, especially building materials. These granites can be utilized in ornamental stones. Therefore, the paper brings new knowledge in the field of the radioactivity of NORM. This was achieved in the manuscript where the authors added the statistical analysis part, which explained and interpreted the presence of high radioactivity in the studied granites.
- Also, the equipment used is not adequate for precise measurements of radioactivity (the usual methodology of measuring radioactivity using gamma-ray spectrometry should be applied, or some other precise measurements.
Response: Before the measurements in the field, the equipment was calibrated using calibration pads according to Moltin et al, (1991). These pads are certified by the international atomic energy agency (IAEA).
- The calculations and conversion of ppm-s to Bq/kg is not adequate.
Response: Corrected and presented in the manuscript.
- Additionally, the results of this study are only off local interest.
Response: The present study is not local because the investigated granites can be exported to different countries of the world. This is an economic project for Egypt, and therefore, this study is considered global.
We thank the Reviewer a lot for the useful and valuable comments that have helped improve the manuscript.
Hoping that all the careful review is sufficient for the direct acceptance of the manuscript, thank you for your time and consideration.
Best wishes,
Mohamed. Y. M. Hanfi
on behalf of all co-authors
Reviewer 3 Report
Authors should include some quantitative results obtained in the abstract section and conclude the abstract properly.
Introduction: Revise the tenses of the manuscript, capital letter cannot start a word in the middle of sentence
2.2 Replace "Radiometric ad mineral analysis" with Radiometric and mineral analysis
Materials and Methods: it is necessary to explain how the portable scintillometer works and how the measurement were obtained.
There is inconsistency in citing references in the manuscript.
Equations should be on a separate line and numbered appropriately. All equations under results and discussion should have been presented under materials and method.
Improve on the quality of the figures
The conclusion should be improved upon to capture the findings of the manuscript.
Author Response
Dear Reviewer,
Please find attached the submission of the carefully revised version of the manuscript in Ref., following the minor comments and modification of the Reviewer.
Below is a detailed list of the changes made in response to the Reviewer’s minor comments (in italics), which outlines every change made a point by point. The changes are marked in the manuscript text (green highlighted).
- Authors should include some quantitative results obtained in the abstract section and conclude the abstract properly.
Modified and re-edited in the manuscript. The abstract was re-written to show the quantitative results and the importance of the present work.
- Introduction: Revise the tenses of the manuscript, capital letter cannot start a word in the middle of sentence
Response: corrected in the manuscript. It was a mistake.
- “2.2 Replace "Radiometric ad mineral analysis" with Radiometric and mineral analysis “
Response: The subtitle changed according to your recommendation.
- Materials and Methods: it is necessary to explain how the portable scintillometer works and how the measurement were obtained.
Response: The application of the portable scintillometer explained in detail in the manuscript.
- There is inconsistency in citing references in the manuscript.
Response: The references are corrected in the manuscript.
- Equations should be on a separate line and numbered appropriately. All equations under results and discussion should have been presented under materials and method.
Response: Modified and the discussion corrected.
- Improve on the quality of the figures
Response: Done in the manuscript. The figures re-plotted.
- The conclusion should be improved upon to capture the findings of the manuscript.
Response: The conclusion re-written to achieve the findings of the manuscript.
We thank the Reviewer a lot for the useful and valuable comments that have helped improve the manuscript.
Hoping that all the careful review is sufficient for the direct acceptance of the manuscript, thank you for your time and consideration.
Best wishes,
Mohamed. Y. M. Hanfi
on behalf of all co-authors
Round 2
Reviewer 2 Report
Dear Editor,
the authors have significantly improved the manuscript in reference to the previous version. The new version is much more appropriate for the special issue of the journal. Hence, I find that the manuscript is now suitable for publication.
However, it must be noted that the English needs significant improvement and extensive editing by a native English speaker!!!